

# Volcanic ash modeling with the NMMB-MONARCH-ASH model: quantification of off-line modeling errors.

Alejandro Marti[1], Arnau Folch[1]

[1]{Barcelona Supercomputing Center (BSC-CNS), Barcelona, Spain}

*Correspondence to*: Alejandro Marti (Alejandro.Marti@bsc.es)

**Abstract.**

Volcanic ash modeling systems are used to simulate the atmospheric dispersion of volcanic ash and to generate forecasts that quantify the impacts from volcanic eruptions on infrastructures, air quality, aviation, and climate. The efficiency of response and mitigation actions is directly associated to the accuracy of the volcanic ash cloud detection and modeling systems. Operational forecasts build on off-line coupled modeling systems where meteorological variables are updated at the specified coupling intervals. Despite the concerns from other communities regarding the accuracy of this strategy, the quantification of the systematic errors and shortcomings associated to the off-line modeling systems has received no attention. This paper employs the NMMB-MONARCH-ASH model to quantify these errors by employing different quantitative and categorical evaluation scores. The skills of the off-line coupling strategy are compared against those from an on-line forecast considered to be the best estimate of the true outcome. Case studies are considered for a synthetic eruption with constant eruption source parameters and for two historical events, which suitably illustrate the severe aviation disruptive effects of European (2010 Eyjafjallajökull) and South-American (2011 Cordón Caulle) volcanic eruptions. Evaluation scores indicate that systematic errors credited by off-line modeling are of the same order of magnitude as those associated to the source term uncertainties. In particular, traditional off-line forecasts employed in operational model setups can result in significant uncertainties, failing to reproduce, in the worst cases, up to 45-70% of the ash cloud of an on-line forecast. These inconsistencies are anticipated to be even more relevant in scenarios where the meteorological conditions change rapidly in time. The outcome of this paper encourages operational groups responsible for real-time advisories for aviation to consider employing computationally efficient on-line dispersal models.





## 1. Introduction

Volcanic ash modeling systems are used to simulate the atmospheric dispersion of volcanic ash and to generate operational short-term forecasts to support civil aviation and emergency management. These systems are vital in efforts to prevent aircraft flying into ash clouds, which could result in catastrophic

impacts (e.g. Miller and Casadevall, 2000; Prata and Tupper, 2009). The aviation community is concerned about the detection and tracking of volcanic ash clouds to provide timely warnings to aircrafts and airports. In the event of an eruption, the individual Volcanic Ash Advisory Center (VAAC) responsible for the affected region combines ash cloud satellite observations and dispersal simulations to issue periodic Volcanic Ash Advisories (VAAs). These are text and graphical products informing on

the extent of the ash clouds at relevant flight levels and their forecasted trajectories at 6, 12 and 18 hours ahead that are updated periodically or whenever significant changes occur in the eruption source term. All this information is used to ensure flight safety by supporting critical decisions such as closure of ash-contaminated air space and airports or diversion of aircraft flight paths to prevent encounters. The noteworthy economic impact and social disruption of these air traffic restrictions are, therefore,

directly associated to the accuracy of the volcanic ash cloud detection and modeling systems.

Volcanic ash modeling systems require of: i) a source term model to characterize the emission of ash depending on the so-called Eruption Source Parameters (ESPs); ii) a meteorological model (MetM) for the description of the atmospheric conditions, and; iii) a Volcanic Ash Transport and Dispersal Model (VATDM; e.g. Folch, 2012) to forecast the particle transport and deposition mechanisms. The MetM

and the VATDM can be coupled either "on-line" or "off-line". In an off-line modeling system, the MetM runs *a priori* and independently from the VATDM to produce the required meteorological fields at regular time intervals, e.g. every 1 or 6 hours for typical mesoscale and global operational MetM outputs respectively. Meteorological fields are then furnished to the VATDM, which commonly assumes constant values for these fields during each time coupling interval or, at most, performs a linear

interpolation in time. This approach is convenient in terms of computing time because different VATDM model executions are possible without re-running the meteorological component, e.g. to update the source term whenever the eruption conditions vary, for inverse modeling of ash emissions (e.g. Marti et al., 2016; Webster et al., 2012), or to perform an ensemble forecast (e.g. Galmarini et al.,



2010) in which all the ensemble members share the same meteorological conditions. However, off-line MetM and VATDM coupling introduces model and numerical errors due to non-synchronized time stepping, use of unaligned grids and projections, and/or inconsistencies in the numerical schemes. In contrast, in an on-line modeling system, the MetM and the VATDM run concurrently and consistently

and the particle transport is automatically tied to the model resolution time and space scales, resulting in a more realistic representation. At present, all operational volcanic ash forecast systems follow the off-line approach and the few existing on-line atmospheric chemistry and transport models specific for volcanic ash, e.g. WRF-Chem (Stuefer et al., 2013) or NMMB-MONARCH-ASH (Marti et al., 2017) are still restricted to a research level. However, notwithstanding the increase of computational power in

recent years, the experiences from other fields (e.g. on-line models for air quality, dust, etc.), and the fact that the total computing time required to run an on-line coupled model is actually not substantially larger (e.g. Grell and Baklanov, 2011; Marti et al., 2017), the benefits of the traditional off-line systems are at question.

Since the 2010 Eyjafjallajökull eruption in Iceland, considerable effort and progress has been done to

quantify and reduce ash cloud modeling and forecasting errors associated with a number of critical aspects (e.g. Bonadonna et al., 2012, 2015) including, among others, characterization of the source term and related uncertainties in model inputs (e.g. Costa et al., 2016) model parameterization of relevant physical phenomena (e.g. aggregation, particle settling velocities, deposition mechanisms, etc.), propagation of errors in the driving MetM forecast, or satellite detection and retrieval algorithms.

However and surprisingly, the quantification of shortcomings associated to the off-line coupling strategy has received no attention despite the fact that lessons from other communities show that these can be substantial (e.g. Baklanov et al., 2014). Errors implicit in off-line coupled systems include inaccurate handling of atmospheric processes occurring at time scales smaller than the coupling interval and eventual feedbacks between the volcanic ash cloud and meteorology. These inconsistencies are

anticipated to be more relevant in scenarios where the meteorological conditions (mainly wind speed and direction) change rapidly in time and for long-range transport simulations due to the model-coupling errors accumulation in time.





The objective of this paper is to quantify the model shortcomings and systematic errors associated with traditional off-line forecasts. In that context, we employ the strategies available in the NMMB-MONARCH-ASH model to evaluate the predictability limitations of the off-line coupling approach against those from an on-line forecast considered to be the best estimate of the true outcome. Section 2

in this manuscript describes the methodology used to quantify the coupling model errors; Section 3 presents the results from a synthetic case study with constant ESPs and focused to quantify the systematic errors attributed to the meteorological coupling intervals. Section 4 evaluates the results from two real cases that suitably illustrate the severe disruptive effects of European (2010 Eyjafjallajökull) and South-American (2011 Cordón Caulle) volcanic eruptions. Section 5 discusses the

magnitude of the model forecast errors implicit in the off-line approach by comparing it with other better-constrained sources of forecast error, e.g. uncertainties in eruption source parameters. Finally, section 6 provides the conclusive remarks of this work.

## 2. Methods

### 2.1. Modeling background

NMMB-MONARCH-ASH (Marti et al., 2017) is a novel on-line meteorological and atmospheric transport model to simulate the emission, transport and deposition of tephra (ash) and aerosol particles released during a volcanic eruption. The model predicts ash cloud trajectories, concentration of ash at relevant flight levels, and the expected ground deposit for both regional and global domains. The on-line coupling in NMMB-MONARCH-ASH allows for solving both meteorology and tephra/aerosol

transport concurrently and interactively at every time-step. The model attempts to pioneer the forecast of volcanic ash and aerosols by embedding a series of new modules on the Barcelona Supercomputing Center (BSC) operational system for short/mid-term chemical weather forecasts (NMMB-MONARCH, formerly known as NMMB/BSC-CTM; Badia et al., 2017; Jorba et al., 2012) developed at the BSC in collaboration with the U.S National Centers for Environmental Prediction (NCEP) and the NASA

Goddard Institute for Space Studies. Its meteorological core, the Non-hydrostatic Multiscale Model on a B grid (NMMB - Janjic and Gall, 2012), is a fully compressible meteorological model with a non-



hydrostatic option that allows for nested global-regional atmospheric simulations by using consistent physics and dynamics formulations. The NMMB model became the North American Mesoscale (NAM) operational meteorological model in October of 2011, and it has been computationally robust, efficient and reliable in operational applications and pre-operational tests since then. In high-resolution

numerical weather prediction applications, the efficiency of the model significantly exceeds those of several established state-of-the-art non-hydrostatic models (e.g. Janjic and Gall, 2012). The computational efficiency of its meteorological core suggests that NMMB-MONARCH-ASH could be used in an operational setting to forecast volcanic ash (Marti et al., 2017).

The model allows for two different coupling strategies: on-line and off-line. The on-line version of the

model runs the MetM and VATDM modules synchronously, updating the transport of ash at each MetM model time step. This coupling strategy offers a more realistic representation of the meteorological conditions, improving the current state-of-the-art of volcanic ash dispersal models, especially in situations where meteorological conditions are changing rapidly in time, two-way feedbacks are significant, or distal ash cloud dispersal simulations are required. In contrast, in the off-line version, the

model uses "effective wind fields" in which, meteorological conditions (e.g. wind velocity, mid-layer pressure, etc.) are set to constant, and are only updated at the user-defined coupling interval. This strategy aims to replicate the decoupling effect of traditional VATDM dispersal models used at operational level.

### 2.2. Forecasts

The skills of an atmospheric dispersal model are known to vary in space and time. In that context, NMMB-MONARCH-ASH simulations are performed to account for the sensibility of the off-line modeling option towards the coupling interval and the dispersal distance of the forecast. On-line forecasts are evaluated against simulations from four different off-line coupling intervals (i.e. 1, 3, 6 and 12h) to compare the skills of each off-line coupling approach. To this purpose, model comparisons are

performed for: i) a synthetic case study with constant ESPs to focus exclusively on the effect of the off-line coupling interval; and ii) two historical cases accounting for the effects of changing the ESPs, including a case where meteorological conditions change rapidly in time (first phase 2011 Cordón



Caulle eruption), and a case where these changes are less abrupt (first phase 2010 Eyjafjallajökull eruption). Finally, in order to assess the order of magnitude of the error associated to the off-line forecasts, we compare it with the better-constrained source of forecast error attributed to the source term (i.e. uncertainties in column height and related mass eruption rate), known to be one of the main reasons

(first order) for VATDM output variability (e.g. Bonadonna et al., 2010).

Forecasts (off and on-line) for each application use the same computational domain and share the same spatial and temporal scales, allowing for a gridded (point-to-point) evaluation. The standard NMMB-MONARCH-ASH parameterization is employed for all simulations (Marti et al., 2017). The

meteorological driver is initialized with wind fields from the Era-Interim reanalysis at 0.75º x 0.75º resolution and, for regional domains, the reanalysis also furnishes 6-h boundary conditions. For the purpose of this study, forecasts predict ash cloud trajectories and concentration of ash at relevant flight levels for a period up to 48 hours. This approach is consistent with most volcanic ash forecasts operational systems.

**2.3. Evaluation methods**

In general terms, forecast evaluation is the process of assessing the goodness of a model prediction. The forecast is compared, or verified, against a corresponding observation of what actually occurred or some good estimate of the true outcome. For the purpose of this work, the output from the on-line forecast is considered to be the model "observations" (i.e. best-estimate of the true outcome) and is compared

against those results from the different off-line forecasts. However, it is important to highlight that the aim of these simulations is not to reconstruct the actual eruptive events but to compare the skills of the off-line forecasts against the on-line in order to quantify their differences.

The accuracy of a volcanic ash forecast can be measured by means of different evaluation scores as no single evaluation score is adequate to fully determine the goodness of a VATDM forecast.

Consequently, a detailed assessment of the strengths and weaknesses of a set of forecasts normally requires more than one or two scores (Jolliffe and Stephenson, 2012). In this study, we evaluate the skills of the off-line versus on-line NMMB-MONARCH-ASH forecasts in terms of their ash column





loading (ACL) using different quantitative and categorical evaluation scores. These scores are often grid-point-based; they compare observations and predictions per grid cell and compute various metrics for the entire set or subset of grid-points. Objects from both on-line and off-line ACL fields must be identified for each evaluation score. An object is a group of adjacent grid cells that have an ash cloud

loading value above a given threshold. Here, the threshold is defined as the typical ash detection limit for most satellite retrievals ($\sim 0.2$ g m$^{-2}$ - Prata and Prata, 2012). Modeled ACL values below this threshold are omitted from all evaluation metrics.

### 2.3.1. Quantitative evaluation scores

Quantitative evaluation scores are useful to determine the degree to which a forecast differs from the
best estimate of the true outcome (i.e. the on-line simulation). Quantitative measures such as correlation coefficients, root-mean-square error (RMSE), or bias, are simple in implementation and thus are regularly used to compare and monitor the quality of a forecast. Here, we use RMSE to assess the average magnitude of forecast errors; bias to assesses the difference between the on-line and off-line forecast means; and the Pearson's correlation coefficient to reflect their linear association. Due to their
invariance properties, these measures are considered to be suitable in many predictive sciences, and in particular in weather and climate forecasting (Jolliffe and Stephenson, 2012).

However, the skill of a dispersion forecast is known to vary in space and time, making these commonly used evaluation scores problematic for grid-point-based measures. A classical example to illustrate these limitations is the "double penalty problem" (Wernli et al., 2008), where a forecast is correct in
terms of amplitude, size and timing, but slightly incorrect concerning location, resulting, for example, in very poorly rated correlation and RMSE scores. To overcome these limitations, we complement the previous scores with the quantitative object-based metric SAL (Wernli et al., 2008). This metric individually considers aspects of the structure (S), amplitude (A), and location (L) of a forecast, revealing meaningful information about the systematic differences between forecasts. This diagnostic
metric has been previously used to measure the skill of volcanic ash forecast using data insertion from satellite observations ( Wilkins et al., 2016) and has been adapted here to compare the quality of on-line and off-line coupled NMMB-MONARCH-ASH forecasts. Figure 1 provides a schematic representation




of different metric combinations and scores in SAL.

The structure [$S$, Eq. (1)] component in SAL captures information about the size and shape of ACL objects by computing the normalized weighted mean mass difference [$V$, Eq. (2)] for the on-line and off-line forecasts:

$$S = \frac{V_{off} - V_{on}}{0.5|V_{off} + V_{on}|} \qquad (1)$$

Weighted means ($V$) of the ash load fields in the column are estimated considering the mass ($R_n$) and the scaled mass [$V_n$, Eq. (3)] for the number of objects in the domain ($M$):

$$V = \frac{\sum_{n=1}^{M} R_n V_n}{\sum_{n=1}^{M} R_n} \qquad (2)$$

Scaled masses ($V_n$) for all objects are calculated separately for each object, as follows:

$$V_n = \sum_{(xy)\in O_n} R_{xy} \Big/ R_n^{max} \qquad (3)$$

where $xy$ is the grid cell location within forecasted field, $R_{xy}$ is the area-integrated concentration field (i.e., ash mass) in grid cell $xy$, and $R_n^{max}$ is the maximum grid cell ash mass in object $O_n$. Note that, in the case of a single object, $V = V_n$. Structure scores range between [-2,2], with positive values

indicating more objects in the off-line forecast and ACL values are too spread out and/or flat. A negative $S$ score occur when the off-line forecast ACL objects cover too small of an area or are too peaked (or a combination of both).

The amplitude ($A$) component corresponds to the normalized difference of the domain-average ash mass

values ($R$). This provides a simple measure of the quantitative accuracy of the total concentration of ash in the domain ignoring the field's subregional structure:




$$A = \frac{\bar{R}_{off} - \bar{R}_{on}}{0.5|\bar{R}_{off} + \bar{R}_{on}|} \tag{4}$$

where $\bar{R}_{off}$ and $\bar{R}_{on}$ are the ash masses averaged over all grid cells in the domain ($D$), i.e., $\bar{R}_n = \sum_{(xy)\in O_n} R_{xy}/D$. Amplitude scores range between [-2,2] with 0 denoting no difference between off-line and on-line forecasts. An amplitude score of +1/-1 indicates that off-line forecasts overestimate/underestimate the domain-averaged ACL by a factor of 3. Scores of $A = 0.4$ and 0.67

correspond to factors of 1.5 and 2, respectively (Wernli et al., 2008).

The location ($L$) component of SAL compares the mass distribution between forecasts. The $L$ component is composed by two parts:

$$L = L_1 + L_2 \tag{5}$$

The first one [$L_1$, Eq. (6)] compares the normalized distance between the center of mass ($C$) of the off-line and on-line ACL fields over the maximum distance within the entire domain ($d$):

$$L_1 = \frac{|C_{off} + C_{on}|}{d} \tag{6}$$

The values of $L_1$ are in the range of [-1,1], with $L_1 = 0$ suggesting identical centers of mass for both forecast. However, separated ash clouds could also have the same center of mass, and therefore $L_1 = 0$ would not necessary indicate a perfect match. The second part of the location component [$L_2$, Eq. (7)] aims at distinguish such situations by measuring the weighted average difference [$H$, Eq. (8)] between the center of mass of the total ash load and the center of mass for each object ($C_n$):

$$L_2 = 2 \left[ \frac{|H_{off} + H_{on}|}{d} \right] \tag{7}$$

$$H = \frac{\sum_{n=1}^{M} R_n |C_{off} + C_{on}|}{\sum_{n=1}^{M} R_n} \tag{8}$$

In the event that both on-line and off-line ACL fields have only one object, then $L_2 = 0$. A factor of 2 is used to scale $L_2$ to the range of $L_1$. Hence, the total location component of $L$ can reach values between





[0,2], and can only be possibly for an off-line forecast where both the distance between objects and the center of mass agree with the on-line forecast. It is important to mention that since both off-line and on-line computational domains are the same, the magnitude dependency of $L$ to the size of the domain does not affect our interpretation of this SAL component.

Absolute SAL scores range from 0 to 6, with scores closest to 0 denoting the best agreement between forecasts. The computation of the structure and location components of SAL requires to group adjacent grid cells into objects with a value above a given threshold for the forecasted variable. For this study, objects are given as $O_n$ , $n = 1, ..., M$, where $M$ is the number of objects in the model domain. Each object combines at least two grid cells to avoid unrealistic single ash-containing grid cells. As defined

previously, the object identification threshold for the ash cloud loading is set to 0.2 g m$^{-2}$. Modeled ACL values below this threshold are omitted from all components of SAL.

### 2.3.2 Categorical evaluation scores

From an operational perspective, it is also important to know whether the presence of volcanic ash constitutes an airspace threat or not. In that context, the significance of quantitative volcanic ash

forecasts can be measured in terms of categorical evaluation scores (Jolliffe and Stephenson, 2012). These scores are less sensitive to larger errors than quantitative evaluations scores. This is particularly important for extremely skewed data such as ACL, providing the degree to which the forecast supports a decision maker during an emergency event (i.e. closure of airspace). Consequently, ash loads can be viewed categorically (or binary for "yes" or "no" events) according to whether that value exceeds a

threshold (event) or not (non-event). Here, we compute a series of categorical evaluation scores based on a contingency table (Table 1), which describes the combined distribution of forecast events and non-events for each coupling strategy.  In Table 1, "Hits" represents the number of grid-points for which both forecasts (off-line and on-line) exceed the threshold previously established (0.2 g m$^{-2}$); "Misses" represents the number of points for which only on-line forecasts exceed this threshold; "False Alarms",

indicates the number of points for which only off-line forecasts exceeded the threshold; Finally, "Correct Negatives", represents the number of points for which either off-line nor on-line forecasts exceeded the threshold value. In this paper, we use these binary skill metrics to calculate four





categorical evaluation scores:

a) *Probability of detection (POD):* Measures the fraction of ash points observed in the on-line forecast and that were correctly predicted for the off-line forecast. This score is good for rare events, should be used together with the $FAR$ score [$FAR$, Eq. (10)], and is insensitive to false alarms. The $POD$
score can reach values between [0,1]:

$$POD = \frac{Hits}{(Hits \ + \ Misses)} \ ; \ [0,1] \tag{9}$$

b) *False alarm ratio (FAR):* Measures the fraction of ash points predicted by the off-line forecast that were observed to be non-events (i.e. non exceeding the threshold) in the on-line forecast. This score should be used together with the previous $POD$ score and ignores the misses. The $FAR$ score can
reach values between [0,1]:

$$FAR = \frac{False \ Alarm}{(Hits \ + \ False \ Alarm)} \ ; \ [0,1] \tag{10}$$

c) *Frequency bias (FBI):* Measures the ratio of frequency of off-line forecast points to the frequency of observed ash points in the on-line forecast. This score indicates whether the forecast system has a tendency to under-forecast ($FBI$<1) or over-forecast ($FBI$>1) events. However, it does not measure
how well the off-line forecast corresponds to the on-line simulation, only measures relative frequencies. The $FBI$ score can reach values between [0,∞]:

$$FBI = \frac{(Hits \ + \ False \ Alarm)}{(Hits \ + \ Misses)} \ ; \ [0, \infty] \tag{11}$$

d) *Critical success rate (CSI):* Measures the fraction of all off-line and on-line forecast points that were correctly diagnosed and does consider both misses and false alarms. The *CSI* score can reach
values between [0,1]:


$$CSI = \frac{Hits}{(Hits + Misses + False\ Alarms)} \; ; \; [0,1] \qquad (12)$$

Similar metrics, such as the Figure of Merit in Space ($FMS$; Galmarini et al., 2010), have been used in previous works (e.g. Wilkins et al., 2016) to complement the SAL score for the evaluation of the spatial coverage between forecasts:

$$FMS = \frac{B_{off} \cap B_{on}}{B_{off} \cup B_{on}} \; ; \; [0,1] \qquad (11)$$

In both cases, a score of 1 suggests a complete spatial overlap of the evaluated forecast. Alternatively, the spatial overlap will decrease as these scores reach values close to 0. Here, we employ the Figure of Merit in Space ($FMS$) metric to evaluate the spatial coverage of the forecasts and to complement a missing spatial coverage component in SAL. To be consistent with our implementation of SAL, the spatial ash coverage is computed only for forecast ACL fields exceeding a threshold of 0.2 g m$^{-2}$. However, it is worth mentioning that a low $FMS$ score could also suggest two similar shapes shifted in space (Mosca et al., 1998) and, therefore, should be used together with the SAL score.

## 3. Synthetic case study

The first step of our evaluation consists in isolating the model's shortcomings and systematic errors that are exclusively associated to the off-line coupling strategy employed in traditional volcanic ash forecasts. To this purpose, we constructed a preliminary synthetic case based on the first 48h of the 2011 Caulle eruption with constant Eruption Source Parameters (ESPs). This synthetic application reduces the differences associated to the source term (i.e. different source term quantification because of different wind fields) and allows us to isolate the systematic errors coming from the off-line coupling approach. Within this framework, we limited the eruption duration to 12h, using a constant column height, and employing the Mastin et al. (2009) relationship (mass eruption rate vs. column height), for the dispersion evaluation of a single bin of ash (1 particle class) during the first 48h of the event. Multiple regional simulations of NMMB-MONARCH-ASH were performed to produce four different



off-line coupled forecasts (i.e. 1, 3, 6 and 12h). Details about the 2011 Caulle eruption, accompanying meteorological conditions, and the computational domain are described in detail in Sect. 4.2. Table 2 provides a summary of the ESPs used for this application. The skills of these forecasts were compared against the on-line coupled simulation employing the quantitative and categorical evaluation scores

described in Section 2.3. Scores at the end of the simulation (48h) are shown in Table 3.  For the purpose of this paper, we focus on describing the scores for the 6h off-line coupled forecast, representative of an operational forecast driven by a global MetM.

Figure 2 shows the results of the quantitative evaluation scores: RMSE (Fig. 2a), correlation coefficient

(Fig. 2b) and bias (Fig. 2c); as a function of the forecast's length for each coupling interval in the synthetic case. These scores assist in determining the degree to which off-line forecasts correspond to the best estimate of the true outcome (on-line forecast). In general terms, and as expected a priori, all scores indicate that the quality of the forecast decreases with longer coupling intervals (i.e. 1, 3, 6 and 12h) and length of the forecast. The RMSE score is presented in Fig. 2a, and is used to assess the

average magnitude of the off-line forecast errors. Figure 2b shows how the linear association between the on-line and off-line forecasts (Pearson's correlation coefficient) significantly decreases with longer coupling intervals, reaching noticeably low correlations. For example, the resulting coefficient for the 6h-coupled forecast after 24h of simulation is below 0.5, and below 0.4 after 48h. These scores indicate that traditional off-line forecasts are not capable to reproduce more than half of the true outcome,

suggesting that the coupling frequency in tephra dispersal modeling could be a critical source of error. This result is relevant considering that 6h-coupled forecasts are used by some emergency model setups. Finally, Figure 2c depicts the forecasts bias over those from the on-line simulation. In general terms, all off-line forecasts underestimate ACL, reaching values between -1 and -5 g m$^2$ at the end of the forecast for the 1h and 12h coupling intervals, respectively (Table 3).

Figure 3 illustrates the results from the quantitative object-based metric SAL, aimed to evaluate the variation in space and time of the forecasts. As with previous scores, the error associated to the SAL score also increases with the length of the coupling frequency. For all off-line simulations within the





synthetic case, the structure component of the metric (Fig 3a) explains most of the discrepancy with the on-line forecast. Negative values of $S$ indicate that off-line forecasts predict fields that cover too small of an area and/or are too peaked. In addition, results from the amplitude and location components indicate a slight overestimation of the domain-averaged ACL for all off-line forecasts, employing

comparable centers of mass with the on-line reference. In general terms, systematic differences in the off-line forecast are 4 times higher for a coupling frequency of 6h than those of 1h interval (Table 3).

Categorical scores resulting from the evaluating of the synthetic case are summarized in Fig. 4. As in the previous scores, a threshold value of 0.2 g m$^{-2}$ is considered to define the ash-contaminated objects,

categorizing these as "yes" or "no" events depending if they exceed or not the threshold. These scores are critical for the aviation industry during a volcanic crisis since they can determine the closure of the airspace or the cancellation of flights.  Figure 4a illustrates the probability of detection ($POD$) for each forecast. As expected, this metric clearly shows how the probability of detecting ash-contaminated points in the off-line forecasts decreases with longer coupling intervals and the forecast length.  In

addition, this figure also suggests that $POD$ scores decrease considerably during the first hours of the forecast, matching the time for which the source was active. This trend is applicable to all categorical evaluation scores. After 48h, the $POD$ scores for the 3h and 6h coupled forecasts are 0.752 and 0.603, respectively. For coupling frequencies above 6h, the probability of detecting an ash-contaminated areas drops below 50% (e.g. 12h couple forecast in Table 3).  $POD$ scores are complemented by the results

from the False Alarm Ratio $(FAR)$ metric, which measures the fraction of ash events predicted by the off-line forecasts that were observed to be non-events. Figure 4b shows $FAR$ scores to be consistent with $POD$ scores and predict a 25% of false ash-contaminated object for the given domain after 48h of the 6h-coupled simulation.  The equivalent plot for the Frequency Bias $(FBI)$ metric as a function of forecast length is shown in Fig. 4c. This metric indicates that all forecasts tend to overestimate the ACL,

especially while the eruption is active. After that time, $FBI$ scores stabilize between values ranging from 0.7 to 1.0.  Finally, Fig. 4d illustrates the spatial overlap between off-line and on-line forecasts defined by the Figure of Merit in Space ($FMS$). This metric provides similar results to the $POD$ metric. However, in this case, false alarms (Table 1) are also considered in the metric leading to $FMS$ scores



lower than those for the *POD* metric. Considering this, *FMS* scores indicate that the spatial overlap (i.e. probability of hits over hits, misses and false alarms) between the on-line and the 6h coupled offline forecasts after 48h of simulation is below 50% (Table 3).

## 4. Application examples

In addition to the synthetic case, we present two applications of NMMB-MONARCH-ASH for the simulation of the initial phases of the 2010 Eyjafjallajökull and 2011 Cordón Caulle eruptions. In these cases, off-line forecasts are evaluated taking into account the effects of the coupling interval and the actual changes in the ESPs (i.e. MER depending on wind field) for each event. A summary of the ESPs used for each application is presented in Table 2. These two events have shed light onto the importance of ash dispersal in the context of aviation safety (Bonadonna et al., 2012), and they suitably illustrate the severe disruptive effects of European and South-American eruptions. Similar to the synthetic case: on-line and off-line forecasts were compared on the same temporal scales and spatial grid; a gridded (point-to-point) evaluation was performed between forecasts following the criteria presented in the contingency Table 1; the output of the on-line forecast was considered as the "observed" (best estimation of true outcome) field; and a threshold of 0.2 g m$^{-2}$ was employed as the ash cloud loading detection limit.

For each application we include: i) a brief description of the eruptive event; ii) a summary of the modeling set-up to simulate the eruption and; iii) a comprehensive evaluation of the plume dispersal forecast including qualitative, quantitative and categorical evaluations and metrics. For the purpose of summarizing the results of these evaluation scores, we focus on describing those scores from the 6h-coupled forecast.

### 4.1. The 2010 Eyjafjallajökull eruption

The April 2010 eruption of Eyjafjallajökull volcano (63.6Nº, 19.6Wº, vent height 1666 m a.s.l.) in southern Iceland, created unprecedented disruptions to European air traffic during 15–20 April. On 14 April a major outbreak of the central crater un- der the covering ice cap lead to a submittal activity



causing phreatomagmatic explosions, generation of volcanic ash, and eruption columns rising up to 9 km (a.s.l) (Institute of Earth Sciences, 2010). The initial ash clouds travelled rapidly across the North Atlantic and North Sea, reaching southern Norway on 15 April and then traveling southwards as a frontal cloud crossing over to many north-European countries. In turn, the London VAAC dispatched

immediate warnings to European aviation authorities and other VAAC centers every 3-6 hours. The southern part of the ash cloud finally grounded in the northern parts of the Alps. On 20 April new guidelines based on safe ash concentration thresholds were adopted, allowing for the ability to resume operations in large areas previously banned. In addition, several other ash cloud episodes occurred during late April and May, disrupting the European airspace for a total of 13 days (over 4 million

passengers stranded due to cancellation or delay of over 100,000 flights), affecting 25 countries, and costing the aviation industry billions of Euros (Oxford Economics, 2010). These impacts brought into focus how significantly volcanoes can affect communities and economies far away from the source, and the critical importance of accurate volcanic ash forecasts.

### 4.1.1. Modeling set-up

For the purpose of simulating this eruption, we employed NMMB-MONARCH-ASH with a model domain consisting of 401x428 grid points, covering the northern and western regions of Europe and using a grid with a horizontal resolution of 0.15º x 0.15º and 60 vertical layers. The top pressure of the model was set to 10 hPa (∼26 km) with a mesh refinement near the top (to capture the dispersion of ash) and the ground (to capture the characteristics of the atmospheric boundary layer). The computational

domain spans in longitude from 30º W to 30º E and in latitude from 34º S to 84º N. The Eruption Source Parameters (ESPs) characterizing the event are described in Table 2 and presented in Fig. 5. Figure 5a shows the variations in column height for the duration of the forecast. Figure 5b illustrates the results from estimating the Mass Eruption Rate (MER) using the different formulations available in NMMB-MONARCH-ASH (Marti et al., 2017). Figure 5c shows the MER variations associated to the

different off-line coupling intervals with the MetM and compare them with the fully coupled on-line forecast.





### 4.1.2 Qualitative evaluation

Figure 6 shows the plume dispersal (ash column loading; ACL) from the on-line forecast corresponding to the first explosive phase (14–18 April) of the Eyjafjallajökull eruption (Gudmundsson et al., 2012). This phase is conveniently divided into 14–16 April, when the volcanic plume produced a well-defined

sector towards the east, and 17 to early 18 April, when northerly winds drove the plume to the south. Complementary to this figure, Figure 7 illustrates the airspace contamination forecasted by the model during the first phase of the eruption at flight levels FL050 and FL200. This figure illustrates the ash hazard aviation guidelines, which distinguish zones of low (green; ash concentration less than 0.2 mg m$^{-3}$), moderate (orange; ash concentration between 0.2 and 2 mg m$^{-3}$) and high (red; ash concentration

above 2 mg m$^{-3}$) concentration of ash employed to regulate No Fly zones. This information is critical for air traffic management to assist flight dispatchers while planning flight paths and designing alternative routes in the presence of a volcanic eruption. Model results show the volcanic cloud traveling E-NE, achieving critical concentration values in northern Europe during 15-17 June, and suggesting severe disruptions in the European airspace.

Figure 8 shows a qualitative comparison between the on-line and the different off-line coupled forecasts for Eyjafjallajökull application. Qualitative comparisons are presented for each coupling interval in different rows (i.e. 1$^{st}$ row = 1h; 2$^{nd}$ row = 3h; 3$^{rd}$ row = 6h; 4$^{th}$ row= 12h coupling). Areas in grey ("Hits") represent grid points for which both forecasts (off-line and on-line) exceed the established

threshold. Red areas ("Misses") indicate those regions where the off-line forecast fails to predict existing ash (underprediction). Finally, blue areas ("False Alarms") illustrate those domain areas for which only off-line forecasts exceed the threshold, implying a false prediction of ash (overprediction). In general terms, off-line forecasts for the Eyjafjallajökull event tend to overpredict towards the north of the plume and to underpredict towards the south. While results of the 1h off-line forecast indicate

mostly Hits (H), Fig. 8 clearly suggests that the number of Missed (M) and False Alarm (FA) points increase with longer coupling intervals and the length of the forecast. This is consistent with those results presented previously in the synthetic case. As a consequence, these forecasts would miss, for example, the arrival of volcanic ash over northern Germany in the late afternoon of 16 April as



indicated by the DWD ceilometer network at the time of the eruption (Flentje et al., 2010). As a general approximation, Fig. 8 suggests that for the Eyjafjallajökull eruption, off-line forecasts with coupling intervals of 3h and above could result in significant inconsistent predictions (M + FA areas).

### 4.1.3. Quantitative and Categorical evaluation

Figure 9 shows the results of the quantitative and categorical metrics for the ACL off-line forecasts for the 2010 Eyjafjallajökull application. Complementing this figure, Table 4 shows the scores for all coupled forecasts after 48h from the eruption starting time. As found in the synthetic case, quantitative and categorical metrics lessen their scores for longer coupling intervals and forecast lengths.

Quantitative evaluation scores RMSE (Fig. 9a), correlation coefficient (Fig. 9b) and bias (Fig. 9c) show

comparable trends than those reported for the synthetic case. After 48h of simulation, the 6h-coupled forecast scores show barely any correlation with the on-line forecast and a RMSE of 0.149 g m$^{-2}$. Bias scores suggest that all off-line forecasts tend to underestimate ACL between -0.33 and -2.5 g m$^2$ at the end of the forecast. Figures 9d through 9g illustrate the results from the quantitative object-based metric SAL, quantifying the variation in space and time of the forecasts. For the Eyjafjallajökull application,

both the structure (Fig. 9d) and amplitude (Fig. 9e) from the off-line forecasts explain most of the discrepancy with the on-line forecast. Contrary to the synthetic case, where the amplitude component (*A*) had a residual effect towards the total SAL, in this application the off-line forecasts tend to underestimate the total concentration of ash in the domain by a factor of 1.5 for coupling intervals equal to or above 6h. This is anticipated since meteorological conditions are kept constant for the given

interval and no additional ash-contaminated objects are found in the domain. After the first coupling with the MetM takes place, scores start to stabilize. After this time, the *A* component stabilizes, reducing its associated underestimation factor. Location scores (Fig. 9f) suggest a comparable mass distribution of the ACL fields for the on-line and off-line forecasts. Finally, absolute SAL scores after 48h of simulations (Table 4) indicate that systematic differences in the off-line forecast are

approximately 2 times higher for a coupling frequency of 6h than those of 1h interval.



Categorical scores for the Eyjafjallajökull application are summarized in Fig. 9h through Fig.9k. Results from the *POD* metric (Fig. 9h) show that the probability of detecting ash-contaminated events in the off-line forecasts decreases with longer coupling intervals, especially during the time the first coupling with the MetM occurs. After 48h, *POD* scores for the 6h-coupled forecast are below 50% (i.e. 0.46).

Conversely, results from the False Alarm Ratio *(FAR)* metric follow an increasing trend (Fig. 9i), misrepresenting near 45% objects in the domain. Results from the Frequency Bias *(FBI)* metric (Fig. 9j) indicate that all off-line forecasts tend to overestimate the ACL. Finally, *FMS* scores suggest that the spatial overlap between the on-line and the 6h coupled offline forecasts after 48h of simulation is below 50% for those simulations with coupling intervals of 3h or more.

**4.2. The 2011 Cordón Caulle eruption**

The 2011 Cordón Caulle eruption exemplifies a typical mid-latitude Central and South Andean eruption. The Cordón Caulle volcanic complex (Chile, 40.5º S, 72.2º W, vent height 1420 m a.s.l.) reawakened on 4 June 2011 around 18:30 UTC after decades of quiescence. The initial explosive phase spanned over more than two weeks, generating ash clouds that dispersed over the Andes (Collini et al.,

2013). The climatic phase (~27 h) (Jay et al., 2014) was associated with a ~9 km (a.s.l.) high column (Osores et al., 2014). For the period between 4-14 June, numerous flights and airports were disrupted in Paraguay, Uruguay, Chile, southern Argentina and Brazil (Wilson et al., 2013).

**4.2.1. Modeling set-up**

The model domain for this application consists of 268x268 grid points covering the northern regions of

Chile and Argentina using a horizontal resolution of 0.15º x 0.15º and 60 vertical layers. The top pressure of the model was set to 10 hPa (~26 km). The computational domain spans in longitude from 41º W to 81º W and in latitude from 18º S to 58º S. The Eruption Source Parameters (ESPs) characterizing the Caulle event are described in Table 2. Figure 10a shows the slight MER variation in column height for the duration of the forecast. Figure 10b illustrates the results from simulating the

MER over time considering different plume models.





### 4.2.2 Qualitative evaluation

Figure 11 illustrates the plume dispersion from the on-line forecast associated to the early Plinian phase (4-7 June) of the Cordón Caulle eruption. The initial ash cloud reached the Atlantic coast on 4 June late afternoon, just before turning to the northeast to reach the northern part of Argentina during the 6 June

and the city of Buenos Aires the days after. The effect of the plume dispersion on air-traffic management is shown in Figure 12. This figure shows the airspace contamination forecasted by the model during 4–6 June at Flight Levels FL050 and FL200. Model results show the volcanic cloud achieving critical concentration values within a wide area east of the Andes range. On 6 June, simulation results show the volcanic cloud moving N-NE, threating the main international airports that

service the province of Buenos Aires. These results suggest that the cancellation of multiple flights in several Argentinean airports during this time was justified.

Figure 13 shows the qualitative comparison between the on-line and off-line coupled forecasts for the first days of 2011 Cordón Caulle eruption. In this case, given that the plume height during the first hours of the eruption was more consistent (no significant changes in wind speed and direction) than for

the Eyjafjallajökull application, the difference between forecasts is less suggestive, although still remarkable. Contrary to the Eyjafjallajökull application, off-line forecasts tended to underestimate to the north of the plume and slightly overestimate to the south. The resulting evaluation from these inconsistencies indicates that off-line forecast with longer coupling intervals missed the abrupt shift in

the plume course known to be associated to early June 6. This alteration was due to the change in the wind direction toward N-NE first and then again towards SE (e.g. Elissondo et al., 2016). As a consequence, these results suggest that off-line forecasts would miss the correct arrival time of volcanic ash to the main international airports in Buenos Aires (i.e. Ezeiza and Aeroparque Jorge Newbery airports).

### 25   4.2.3. Quantitative and Categorical evaluation

Figure 14 summarizes the results for the quantitative and categorical metrics for the 2011 Caulle application. Metric scores at the end of the simulation are presented in Table 5. Results for the Cordón





Caulle application are consistent with those from the synthetic case and the Eyjafjallajökull application in that the uncertainty of the forecast increases significantly with the length of the coupling frequency employed. Quantitative evaluation scores RMSE (Fig. 14a), correlation coefficient (Fig. 14b) and bias (Fig. 14c) show comparable trends to those from the Eyjafjallajökull application. Despite this similarity,

linear correlation coefficients between off-line and on-line forecasts for the Caulle application are higher than those from the Eyjafjallajökull simulation. This result is explained by the fewer changes in the source term (e.g. variations in the column height) during the Caulle event. After 48h of simulation, the 6h-coupled forecast scores a correlation coefficient of 0.60 with a RMSE of 0.16 g m$^{-2}$. Bias scores suggest that all off-line forecasts tend to underestimate ACL between -0.13 and -4.75 g m$^{-2}$ at the end of

the forecast.

Figures 14d through 14g illustrate the results from the quantitative object-based metric SAL for the Cordón Caulle event. SAL scores (Fig. 14g) suggest that differences between on-line and off-line strategies for the Cordón Caulle application are considerably higher than those for the Eyjafjallajökull application. This is due to the changing meteorological conditions during to the Cordón Caulle event,

and confirms that inconsistences associated to off-line forecasts are more relevant in scenarios where the meteorological conditions (mainly wind speed and direction) vary rapidly in time. In terms of the individual components of SAL, Structure (Fig. 14d) and Amplitude (Fig, 14e) scores explain most of the discrepancy with the on-line forecast. Structure scores indicate that more objects occur in the off-line forecast and ACL values are too spread out and/or flat, while Amplitude scores suggest that off-line

forecasts tend to overestimate the total concentration of ash in the domain up to a 1.5 factor. The systematic error associated to the off-line forecasts is clearly demonstrated after 18h of simulation (Fig. 14g), time during which changes in wind speed and direction start to be noteworthy (Fig. 11). Location scores (Fig. 14f) suggest a consistent mass distribution of the ACL fields amongst forecasts. The Cordón Caulle application is a perfect example to illustrate the importance of complementing traditional

quantitative metrics with the quantitative object-based metric SAL. For this particular case, SAL scores are capable to capture the inconsistences of the off-line dispersion forecast due to the changing meteorological conditions that other quantitative metrics (i.e. RMSE, correlation coefficient, bias) cannot account for.





Finally, categorical scores for the Cordón Caulle application are presented in Fig. 14h through Fig. 14k. Results suggest that the skill of the forecast decreases with longer coupling intervals, following similar trends than those found in the Eyjafjallajökull application. After 48h, $FMS$ scores suggest that the spatial overlap between the on-line and the 6h coupled offline forecasts is below 65% (Table 5; Fig 14k), with a probability of misrepresenting ash-contaminated objects above 10% ($FAR$; Fig. 14i). Results from the Frequency Bias *(FBI)* metric (Fig. 14j) indicate that all off-line forecasts tend to overestimate the ACL.

## 5. Discussion

Volcanic ash modeling systems are used to simulate the atmospheric dispersion of volcanic ash and to generate forecasts to quantify the impacts from volcanic eruptions on air quality, aviation, and climate. However, volcanic ash forecasts require the consideration of numerous and complex uncertainties. The 2010 Eyjafjallajökull eruption clearly demonstrated the need for a better understanding of the uncertainties associated to the dispersal model employed in operational volcanic ash forecasting. Since then, the scientific community has focused on identifying and improving uncertainties primarily associated to the characterization of the source term (e.g. MER, column height, etc.). However and surprisingly, the quantification of systematic errors and shortcomings associated to the meteorological data driving the dispersion model has received little attention. Traditionally, operational volcanic ash forecasts employ off-line coupling strategies to produce the required meteorological fields at regular time intervals, e.g. every 1 or 6 hours for typical mesoscale and global operational MetM configurations respectively. In previous sections of this paper, we have shown the meaningful negative impact of employing off-line coupling intervals on the accuracy of the ash-cloud simulations as compared to on-line coupled forecasts. In particular, Section 3 showed the scores from evaluating a synthetic case focusing exclusively on the effect of the coupling approach. Evaluation scores reveal that the uncertainty of the off-line forecasts increase significantly with the length at which the meteorological driver is coupled with the dispersion model (e.g. up to 4 times for the 6h-coupled forecast). However,





the question on how does this compares to other better-constrained sources of forecast error remains still unanswered.

This section aims to answer this question by evaluating to which extend the magnitude of the model forecast errors implicit in the offline approach compares with that of the source term. To this purpose,
we performed four additional experimental simulations under the synthetic case where ESP for the online forecast where modified by: i) employing a ± 2 MER factor (i.e. x2 and 1/2 times the original MER), and; ii) varying ± 20% the corresponding column height. As in previous applications, experimental forecasts were evaluated on the same temporal scales and spatial grid against on-line forecast employing a range of complementary quantitative and categorical metrics. Figure 15 illustrates
the evaluation scores from these four simulations, and compares them with those of the 6h coupled offline forecast. Overall, Fig. 15 reveals that systematic errors and shortcomings associated to the traditional off-line coupling strategies employed in operational volcanic ash forecast are of the same other of magnitude than those uncertainties credited by the characterization of the source term. For example, correlation coefficients (Fig. 15b) and *POD* scores (Fig. 15h) suggest an additional 10-30%
level of uncertainty attributed to the 6h coupled forecast than those associated to the source term. In that same context, at the end of the simulation, *FMS* scores (Fig. 15k) reveal that the spatial overlap between the on-line and the 6h coupled offline forecasts is ∼ 20% lower than those from varying ± 20% the column height, and ∼ 50% lower than those from altering the original MER.

These results suppose a significant advance in the quantification of the uncertainty sources associated to
traditional off-line volcanic ash forecasts.

## 6. Conclusions

This paper quantifies the systematic errors inherent in off-line coupling strategies employed for operational volcanic ash forecasting. To this purpose, we employed the different coupling strategies available in the NMMB-MONARCH-ASH model to evaluate the predictability limitations of the off-
line forecast against the on-line. Model comparison were performed for a synthetic case study focusing exclusively on the effect of the coupling approach, and for two historical cases accounting for changing





meteorological conditions and ESPs. Evaluation scores indicate that systematic errors credited by off-line forecast are of the same order of magnitude as those better-constrained uncertainties associated to the source term. In particular, off-line forecasts in operational setups can result in significant errors in the dispersion of the ash plume for coupling intervals above 3h. The results of this study show that 6h

coupling off-line forecasts fail to reproduce over 50% of the on-line forecast (best estimate of the true outcome) for a case with constant ESPs (synthetic case); close to 70% for the 2010 Eyjafjallajökull case and over 45% for the 2011 Cordón Caulle case. These inconsistencies are anticipated to be even more relevant in scenarios where the meteorological conditions change rapidly in time. The outcome of this paper advocates that operational groups responsible for real-time advisories for aviation consider

employing computationally efficient on-line dispersal models.

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





**TABLES**

| Off-line forecast exceeding threshold | On-line forecast exceeding threshold | |
|---|---|---|
| | Yes | No |
| Yes | *Hits* | *False Alarm* |
| No | *Misses* | *Correct Negatives* |

**Table 1: Contingency table of binary events for categorical verifications scores at each grid-point.**

| Source term | Synthetic | 2010 Eyjafjallajökull | 2011 Cordón Caulle |
|---|---|---|---|
| Run duration | 12h | 96h | 72h |
| Vertical distribution of mass in the column | Suzuki (1983) distribution | Suzuki (1983) distribution | Suzuki (1983) distribution |
| MER vs. column height relationship | Mastin et al. (2009) | Degruyter and Bonadonna (2012) (Fig. 5b) | Degruyter and Bonadonna (2012) (Fig. 10a) |
| Column height | 8500m | Fig. 5c | Fig. 10b |
| TGSD | 1 bin (Φ=6) | (Bonadonna et al., 2011) | Bonadonna et al., (2015b) |
| Sedimentation model | (Ganser, 1993) | (Ganser, 1993) | (Ganser, 1993) |

**Table 2: Summary of Eruption Source Parameters (ESPs) used in NMMB-MONARCH-ASH for the synthetic case, and the 2010 Eyjafjallajökull and 2011 Cordón Caulle applications.**

| Coupling/Score | R | RMSE | BIAS | S | A | L | |SAL| | POD | FAR | FBI | FMS |
|---|---|---|---|---|---|---|---|---|---|---|---|
| 1h | 0.849 | 0.107 | -1.090 | -0.026 | -0.007 | 0.006 | 0.039 | 0.897 | 0.039 | 0.934 | 0.855 |
| 3h | 0.631 | 0.167 | -2.589 | -0.077 | -0.008 | 0.012 | 0.097 | 0.752 | 0.108 | 0.843 | 0.669 |
| 6h | 0.357 | 0.220 | -3.362 | -0.143 | 0.008 | 0.027 | 0.178 | 0.603 | 0.243 | 0.796 | 0.477 |
| 12h | 0.039 | 0.269 | -5.239 | -0.077 | -0.003 | 0.027 | 0.107 | 0.400 | 0.413 | 0.682 | 0.281 |

5    **Table 3, Evaluation scores for the synthetic case at the end of the 48h forecast with NMMB-MONARCH-ASH.**





| Coupling/Score | R | RMSE | BIAS | S | A | L | \|SAL\| | POD | FAR | FBI | FMS |
|---|---|---|---|---|---|---|---|---|---|---|---|
| 1h | 0.787 | 0.072 | -0.327 | -0.087 | -0.019 | 0.006 | 0.112 | 0.881 | 0.091 | 0.969 | 0.805 |
| 3h | 0.386 | 0.122 | -0.783 | -0.138 | -0.049 | 0.019 | 0.206 | 0.664 | 0.283 | 0.926 | 0.499 |
| 6h | 0.08 | 0.149 | -1.82 | -0.116 | -0.063 | 0.034 | 0.213 | 0.465 | 0.438 | 0.828 | 0.332 |
| 12h | -0.292 | 0.177 | -2.521 | -0.136 | -0.105 | 0.051 | 0.292 | 0.251 | 0.671 | 0.762 | 0.156 |

**Table 4. Evaluation scores for the 2010 Eyjafjallajökull eruption application at the end of the 48h forecast with NMMB-MONARCH-ASH.**

| Coupling/Score | R | RMSE | BIAS | S | A | L | \|SAL\| | POD | FAR | FBI | FMS |
|---|---|---|---|---|---|---|---|---|---|---|---|
| 1h | 0.932 | 0.068 | -0.131 | -0.001 | -0.02 | 0.002 | 0.023 | 0.965 | 0.028 | 0.993 | 0.934 |
| 3h | 0.808 | 0.113 | -1.16 | -0.008 | -0.038 | 0.008 | 0.054 | 0.881 | 0.063 | 0.94 | 0.82 |
| 6h | 0.598 | 0.164 | -3.657 | 0.123 | -0.105 | 0.017 | 0.245 | 0.719 | 0.114 | 0.811 | 0.639 |
| 12h | 0.333 | 0.212 | -4.754 | 0.033 | -0.287 | 0.046 | 0.366 | 0.568 | 0.248 | 0.755 | 0.453 |

**Table 5. Evaluation scores for the 2011 Cordón Caulle eruption application at the end of the 48h forecast with NMMB-MONARCH-ASH.**





**FIGURES**

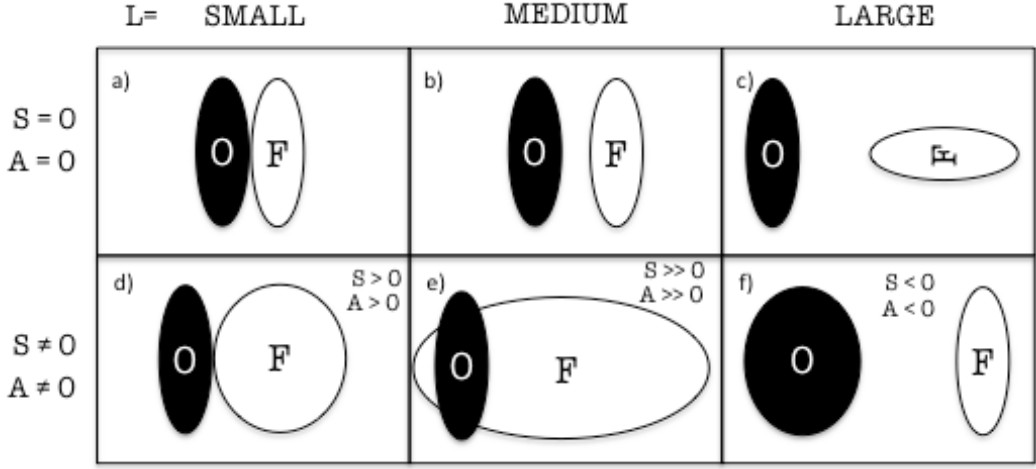

**Figure 1. Schematic representation of the possible on-line (O; representing the "observations") and off-line forecasts (F) combinations of the different components for the quantitative object-based metric SAL: Structure (S), Amplitude (A) and Location (L). Modified from *Wernli et al.* (2008).**



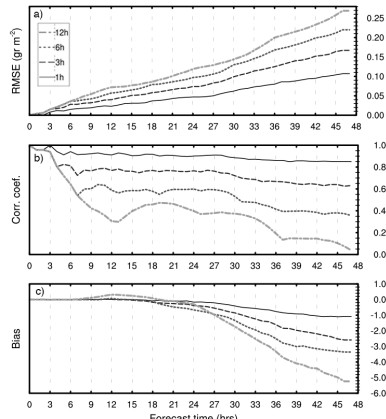

**Figure 2.** Quantitative evaluation scores for NMMB-MONARCH-ASH synthetic application: a) root mean square error; b) Pearson's correlation coefficient; c) error bias.

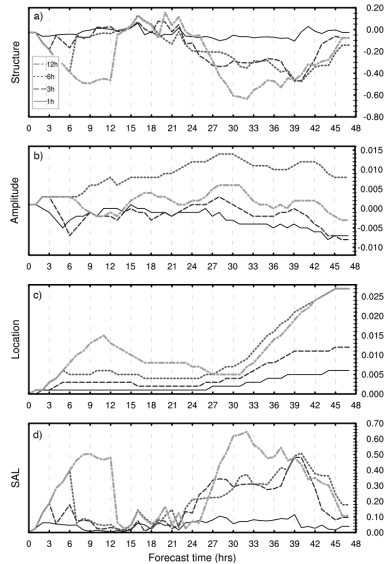

**Figure 3.** SAL evaluation scores for NMMB-MONARCH-ASH synthetic case: a) Structure; b) Amplitude; c) Location; d) combined SAL.



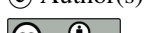

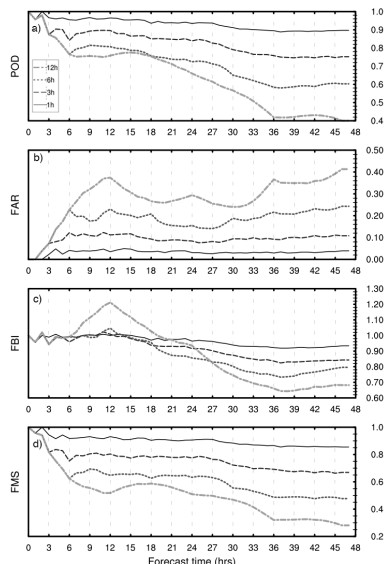

**Figure 4. Categorical evaluation scores for NMMB-MONARCH-ASH synthetic case including: a) Probability of detection (POD); b) False alarm ratio (*FAR*); c) Frequency bias (FBI), and; d) Figure of Merit in Space (FMS).**





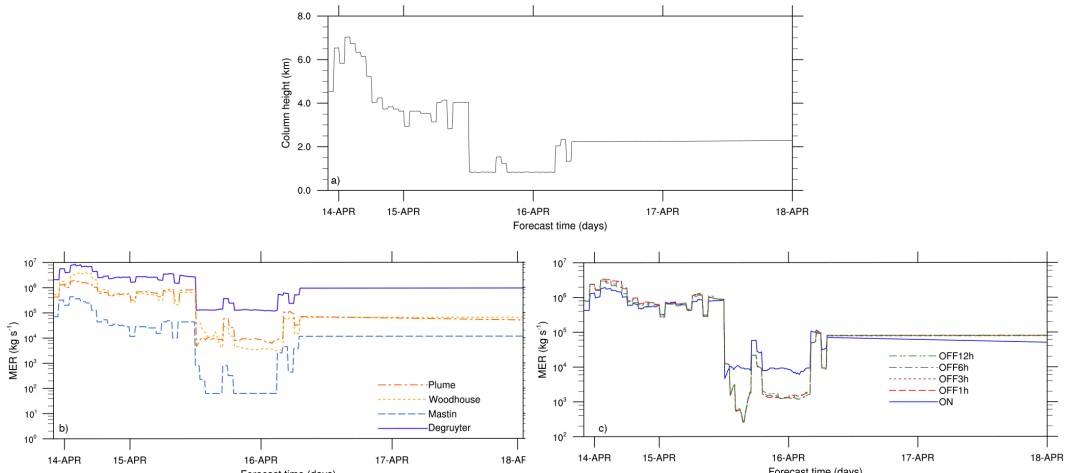

**Figure 5. Eruption Source Parameters for the 2010 Eyjafjallajökull application: a) Column height fluctuation over time; b) Resulting MER over time considering different parameterizations; c) Resulting MER for each coupling strategy (Degruyter option only).**

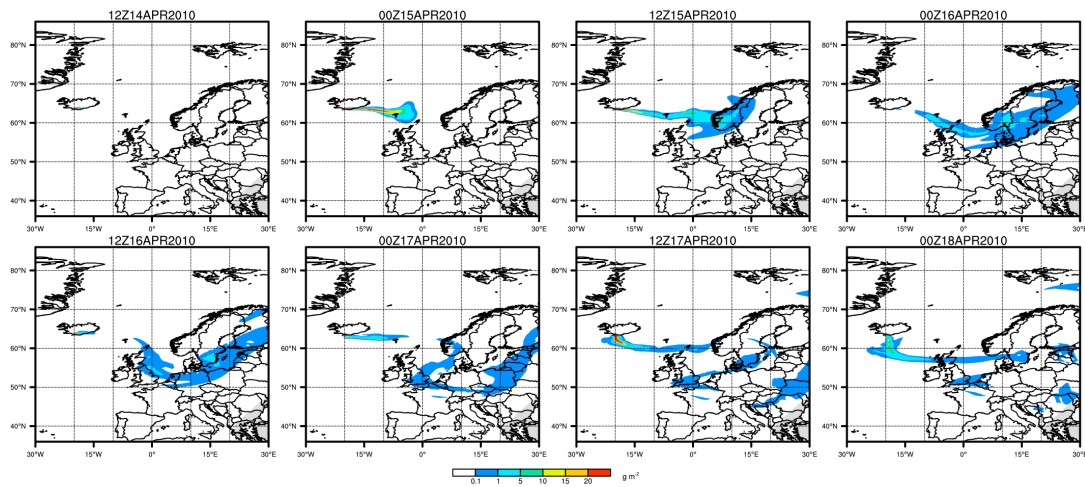

**Figure 6. NMMB-MONARCH-ASH total ACL (mass loading; g m⁻²) for the 2010 Eyjafjallajökull application.**





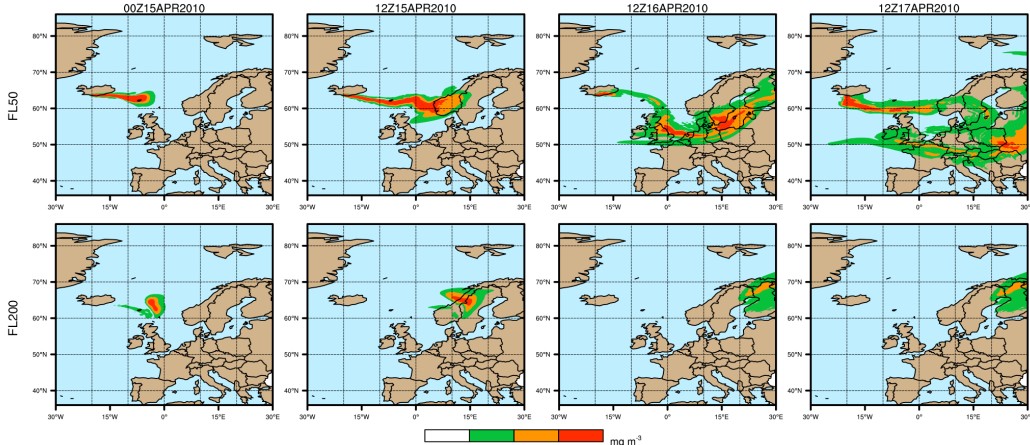

**Figure 7. Ash hazard aviation guidelines applied for 2010 Eyjafjallajökull application over time. Zones of low (green; ash concentration <0.2 mg m$^{-3}$), moderate (orange; ash concentration between 0.2 and 2 mg m$^{-3}$) and high (red; ash concentration above 2 mg m$^{-3}$) concentration are displayed for FL50 (top) and FL200 (bottom).**

5   .





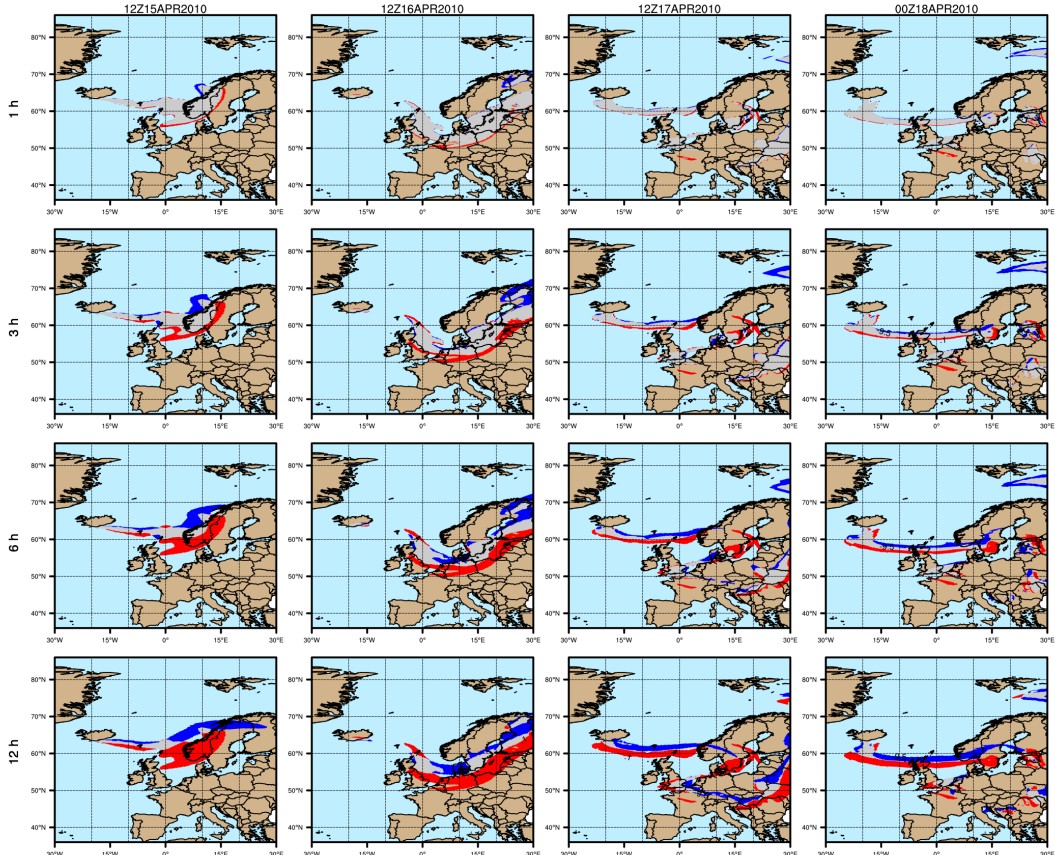

**Figure 8. Qualitative comparison between the on-line and off-line forecasts with 1h (row 1), 3h (row 2), 6h (row 3) and 12h (row 4) coupling intervals. Gridded evaluation is performed following the criteria presented in the contingency Table 1. Hit (grey), Missed (red) and False Alarm (blue) predictions are shown for the 2010 Eyjafjallajökull case over time.**





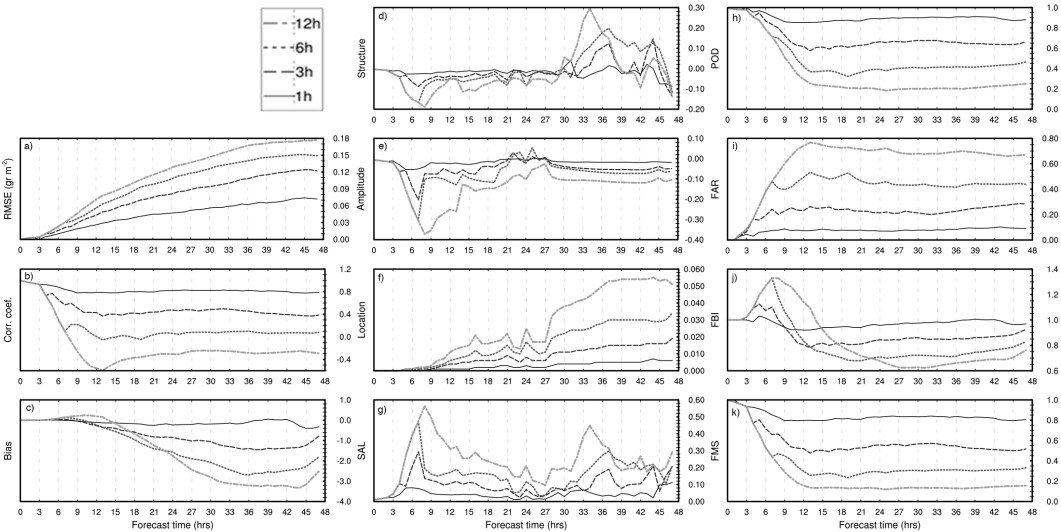

**Figure 9. On-line vs. off-line evaluation scores for the 2010 Eyjafjallajökull case.**

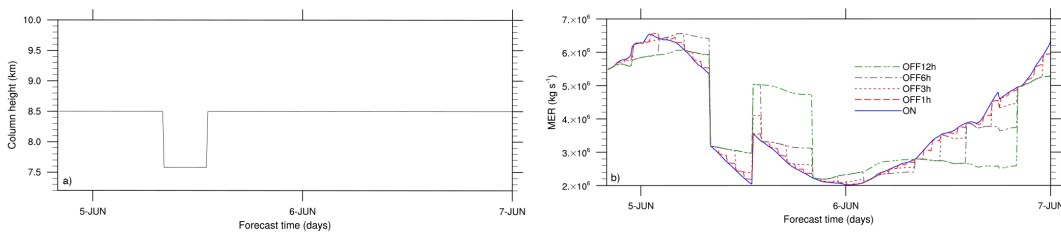

5    **Figure 10. Eruption source parameters for the 2011 Cordón Caulle case: a) Column height fluctuation over time.; b) Resulting MER over time for each coupling strategy**




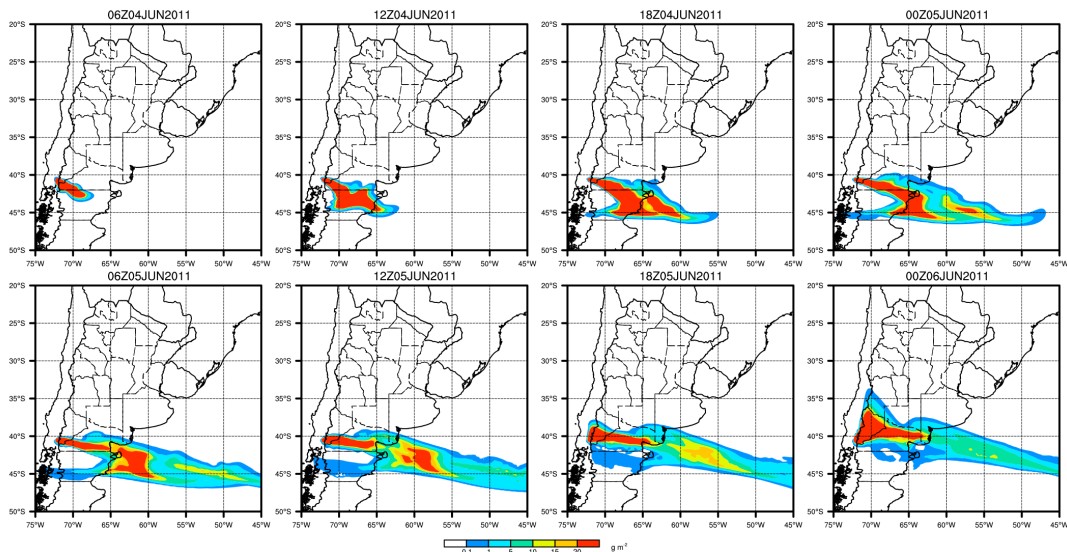

**Figure 11. NMMB-MONARCH-ASH total column load (mass loading; g m⁻²) for the 2011 Cordón Caulle case.**



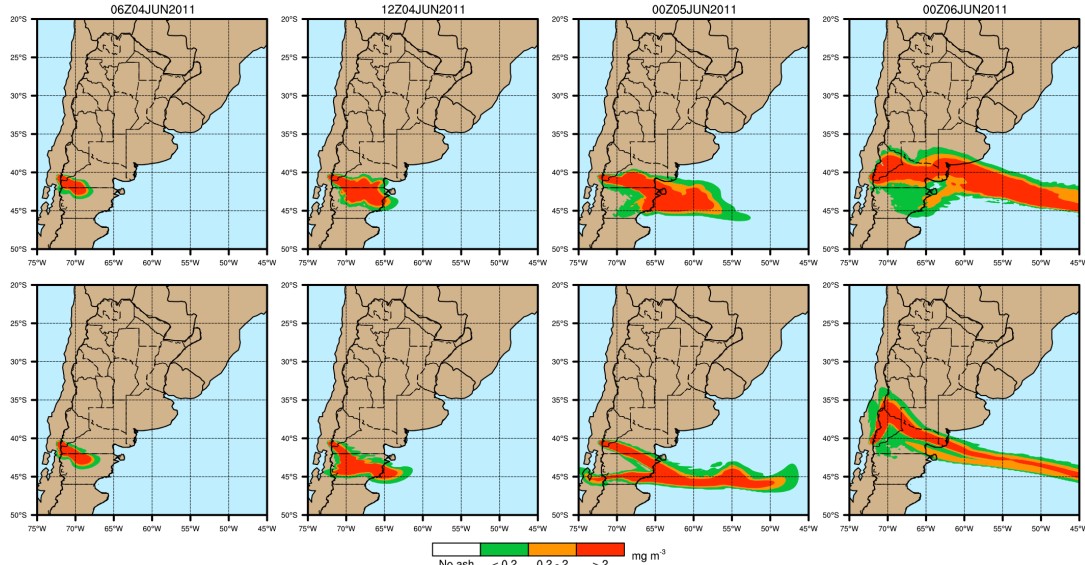

**Figure 12. Ash hazard aviation guidelines applied for the 2011 Cordón Caulle application over time. Zones of low (green; ash concentration <0.2 mg m$^{-3}$), moderate (orange; ash concentration between 0.2 and 2 mg m$^{-3}$) and high (red; ash concentration above 2 mg m$^{-3}$) concentration are displayed for FL50 (top) and FL200 (bottom).**





.

**Figure 13. Qualitative off-line vs. on-line forecast comparison for 1h (row 1), 3h (row 2), 6h (row 3) and 12h (row 4) coupling intervals. Gridded evaluation is performed following the criteria presented in the contingency Table 2. Hit (grey), Missed (red) and False Alarm (blue) predictions are shown for the 2011 Cordón Caulle application over time.**





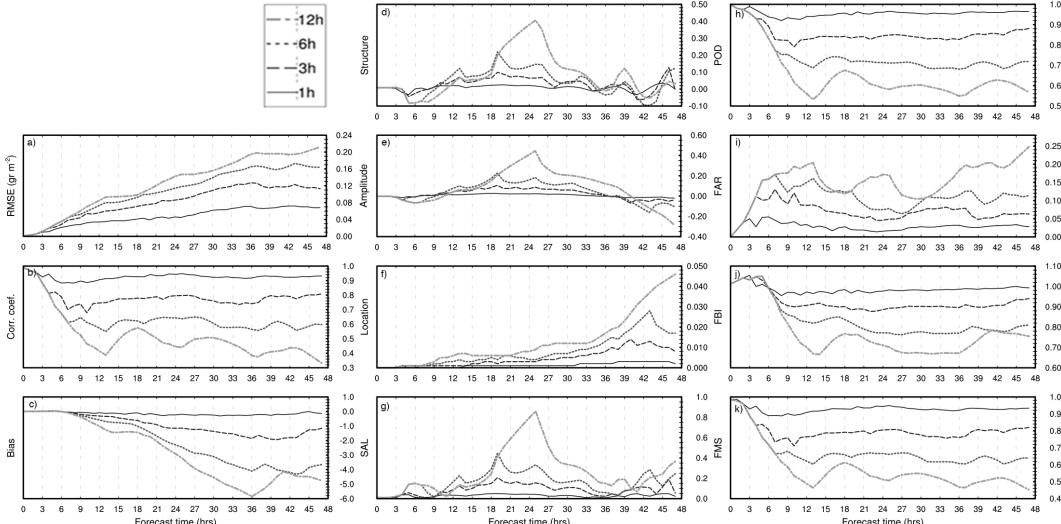

**Figure 14.** On-line vs. off-line evaluation scores for the 2011 Cordón Caulle application.





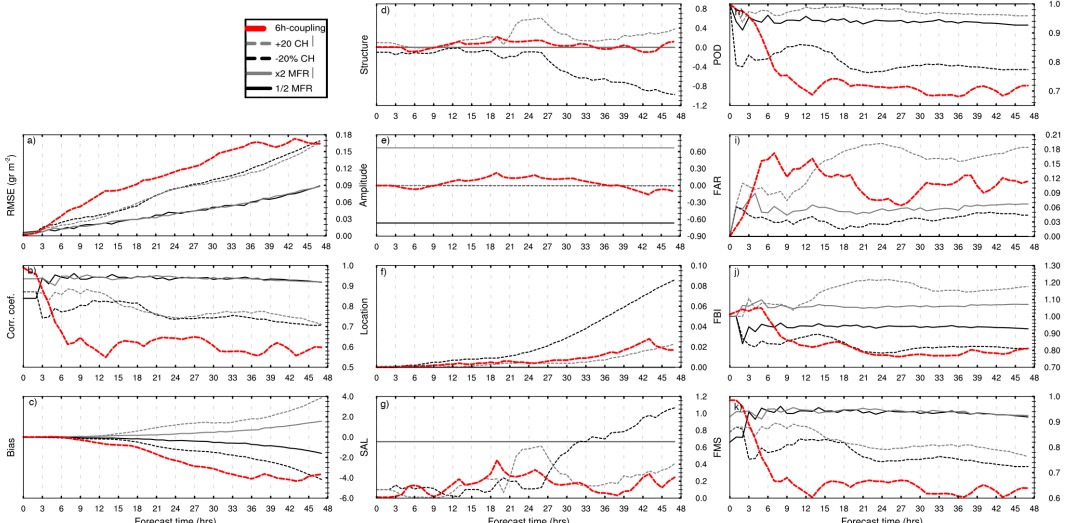

**Figure 15. On-line vs. off-line evaluation scores evaluation scores for the NMMB-MONARCH-ASH synthetic application representing the uncertainty associated to the source term. ESPs were modified for the eruption column height (+/- 20%) and MER (x2 and ½). Scores are compared with those from the 6h off-line coupled forecasts (red line).**

