# Peer review of "Volcanic ash modeling with the NMMB-MONARCH-ASH model: quantification of off-line modeling errors."

_Atmospheric Chemistry and Physics, 2017_

## Referee Comment (RC1) · Anonymous Referee #2 · 20 Jun 2017

I recommend the paper:

**Volcanic ash modeling with the NMMB-MONARCH-ASH model: Quantification of off-line modeling errors by Alejandro Marti and Arnau Folch**

for publication in Atmospheric Chemistry and Physics after a minor revision.

**General Comments**

1. The paper adresses issues that are highly relevant for modeling atmospheric transport of aerosols and particulate matter with the emphasis on volcanic ash. In particular, the off-line (Lagrangean trajectory based) and the on-line (Eulerian) approaches are compared and the accuracy of the off-line method is assessed and quantified versus that of the on-line one. In this way excellent data points are obtained for choosing the most adequate approach for research and operational applications. Such a study has been long overdue.

2. The scientific methods and assumptions are valid and clearly stated. The results of the comparisons are illustrated by an idealized and two well designed and thoroughly described and examined real data test cases. The discussion and conclusions are well supported by the data.

3. The title is adequate, and the abstract gives a concise and clear summary. The presentation structure and the language are generally very good, but some minor polishing at few places would be benefitial.

**Specific comments**

1. p. 2, line 30-31

There are on-line operational systems, e.g., for dust transport.

2. p. 10, line 15 and elsewhere in the manuscript

The prase "... decreases with coupling frequency..." may be misunderstood to mean the opposite of what is actualy meant, i.e. "... decreases as the coupling frequency increases ..." Therefore, I would recommend to say "... decreases with decreasing coupling frequency ..." instead.

**Suggestions for technical corrections**

1. p. 1, line 20 Replace "credited to" by "due to" 2. p. 1, line 20 Replace "that" by "as"

3. p. 2, line 14 and elswere in the text Leave out "of" in "require of"

---

## Referee Comment (RC2) · L. Mastin (Referee) · 20 Jun 2017

Comments to the paper: "Volcanic ash modeling with the NMMB-MONARCH-ASH model: quantification of off-line modeling errors" By Alejandro Marti and Arnau Folch This paper compares results of volcanic ash-cloud simulations generated with the on-line model NMMB-MONARCH-ASH with those of the same model, used offline. The on-line mode calculates both meteorology and ash dispersion simultaneously whereas the offline mode reads pre-calculated meteorology at designated time (or "coupling") intervals. The manuscript compared on-line and offline results for three cases: a synthetic eruption and the real eruptions of Eyjafjallajökull (2010) and Cordon Caulle

(2011). The comparison found large differences in cloud location and concentration, which increased with time during a single simulation and which were greater as the coupling interval increased, from 1 to 12 hours. This study pointed out a critical and perhaps poorly appreciated advantage of on-line modeling; that coupling meteorology with ash dispersal models run offline can lead to significant errors if met. conditions are changing rapidly. Prior to reading this manuscript I had assumed that on-line models were an advantage primarily in rare cases where the ash cloud influenced wind or other meteorological conditions. The paper is written clearly, the results appear to be robust, and I think are highly significant. Therefore I highly recommend publication. There are however some minor changes that I think would help the paper before publication 1) How does the offline version of your model handle meteorology at the times between the coupling times? On p. 5, line 16, you suggest that meteorological parameters are set to constant in between coupling times. But many offline models linearly interpolate. Could higher-order interpolation schemes reduce error? 2) Section 2.3.1. It was difficult for me to grasp the physical significance of some of the quantities used to compare output from the offline and online models. For example, the Structure component S is said (line 3, page 8) to capture information about the size and shape of cloud objects. But all of the terms in S refer to mass; of a node, column or nodes, or cloud object. How do differences in S reflect variations in size and/or shape? The exact meaning of some other parameters, such as $R_{xy}$ and D were unclear. And it was unclear to me how one could get a value less than zero for $L_1$. Also, I didn't see a definition of the parameter B in eq. 11. More comments are included in the returned pdf. 3) Several source parameter terms in Table 2 are not adequately explained. Details are in comments in that table. Also, I don't see sources cited for the observations used to constrain the eruption source parameters. Other key observations, like the arrival time of the Cordon Caulle ash cloud in Buenos Aires on 4 June 2011 (p. 20, line 3) do not cite sources. 4) Some of the figures need more description. For example, the methods of estimating mass eruption rate plotted in Fig. 5b. And the various lines in Fig. 10b representing mass eruption rate with time. 5) Section 4.1.2: In examining the

[Figure]

Eyjafjallajökull eruption, why do you think the misses were in the south and the false alarms were in the north? Dacre et al. (2011) suggested there was an error in the modeled wind speed over England. Was the south-oriented wind just accelerating, and the acceleration was not being caught when the coupling intervals were too infrequent? 6) Section 4.1.3: You say that bias scores suggest that offline forecasts tend to systematically underestimate ash column loading? Is there a physical explanation for this? If the model is conserving mass, does this imply that offline models also systematically overestimate cloud area? And how does this statement square with your statement on p. 19, line 6, that all off-line forecasts OVER-estimate ash column loading?

Additional minor and specific comments are included in the attached pdf. I look forward to seeing the final version of this paper.

Larry Mastin

References: Dacre, H. F., et al. (2011), Evaluating the structure and magnitude of the ash plume during the initial phase of the 2010 Eyjafjallajökull eruption using lidar observations and NAME simulations, Journal of Geophysical Research: Atmospheres, 116(D20), n/a-n/a, doi:10.1029/2011JD015608.

Please also note the supplement to this comment:
http://www.atmos-chem-phys-discuss.net/acp-2017-354/acp-2017-354-RC2-supplement.pdf

———————————————————

[Figure]

**Supplement:**

[revised manuscript text omitted]
{\overline{R}_{off} - \overline{R}_{on}}{0.5|\overline{R}_{off} + \overline{R}_{on}|}$$

(4)

where  $\overline{R}_{off}$  and  $\overline{R}_{on}$  are the ash masses averaged over all grid cells in the domain (D), i.e.,  $\overline{R}_n = \sum_{(xy)\in O_n} \frac{R_{xy}}{D}$ . Amplitude scores range between [-2,2] with 0 denoting no difference between off-line and on-line forecasts. An amplitude score of  $+1\frac{1}{L}1$  indicates that off-line forecasts overestimate/underestimate the domain-averaged ACL by a factor of 3. Scores of A = 0.4 and 0.67

5 correspond to factors of 1.5 and 2, respectively (Wernli et al., 2008).

The location (L) component of SAL compares the mass distribution between forecasts. The L component is composed by two parts:

$$L = L_1 + L_2 \tag{5}$$

The first one  $[L_1, \text{Eq.} (6)]$  compares the normalized distance between the center of mass (C) of the offline and on-line ACL fields over the maximum distance within the entire domain (d):

$$L_1 = \frac{|C_{off} + C_{on}|}{d} \tag{6}$$

10 The values of  $L_1$  are in the range of [-1,1], with  $L_1 = 0$  suggesting identical centers of mass for both forecast. However, separated ash clouds could also have the same center of mass, and therefore  $L_1 = 0$ would not necessary indicate a perfect match. The second part of the location component  $[L_2, \text{Eq. (7)}]$ aims at distinguish such situations by measuring the weighted average difference [H, Eq. (8)] between the center of mass of the total ash load and the center of mass for each object  $(C_n)$ :

$$L_2 = 2\left[\frac{\left|H_{off} + H_{on}\right|}{d}\right] \tag{7}$$

$$H = \frac{\sum_{n=1}^{M} R_n |C_{off} + C_{on}|}{\sum_{n=1}^{M} R_n}$$
(8)

15 In the event that both on-line and off-line ACL fields have only one object, then  $L_2 = 0$ . A factor of 2 is used to scale  $L_2$  to the range of  $L_1$ . Hence, the total location component of L can reach values between

[0,2], and can only be possibly for an off-line forecast where both the distance between objects and the center of mass agree with the on-line forecast. It is important to mention that since both off-line and online computational domains are the same, the magnitude dependency of *L* to the size of the domain does not affect our interpretation of this SAL component.

- 5 Absolute SAL scores range from 0 to 6, with scores closest to 0 denoting the best agreement between forecasts. The computation of the structure and location components of SAL requires to group adjacent grid cells into objects with a value above a given threshold for the forecasted variable. For this study, objects are given as  $O_n$ , n = 1, ..., M, where M is the number of objects in the model domain. Each object combines at least two grid cells to avoid unrealistic single ash-containing grid cells. As defined
- 10 previously, the object identification threshold for the ash cloud loading is set to 0.2 g m-2. Modeled ACL values below this threshold are omitted from all components of SAL.

**2.3.2 Categorical evaluation scores**

From an operational perspective, it is also important to know whether the presence of volcanic ash constitutes an airspace threat or not. In that context, the significance of quantitative volcanic ash

- 15 forecasts can be measured in terms of categorical evaluation scores (Jolliffe and Stephenson, 2012). These scores are less sensitive to larger errors than quantitative evaluations scores. This is particularly important for extremely skewed data such as ACL, providing the degree to which the forecast supports a decision maker during an emergency event (i.e. closure of airspace). Consequently, ash loads can be viewed categorically (or binary for "yes" or "no" events) according to whether that value exceeds a
- 20 threshold (event) or not (non-event). Here, we compute a series of categorical evaluation scores based on a contingency table (Table 1), which describes the combined distribution of forecast events and nonevents for each coupling strategy. In Table 1, "Hits" represents the number of grid-points for which both forecasts (off-line and on-line) exceed the threshold previously established (0.2 g m-2); "Misses" represents the number of points for which only on-line forecasts exceed this threshold; "False Alarms",
- 25 indicates the number of points for which only off-line forecasts exceeded the threshold; Finally, "Correct Negatives", represents the number of points for which either off-line nor on-line forecasts exceeded the threshold value. In this paper, we use these binary skill metrics to calculate four

---

## Author Response (AR1)

Dear Dr. Galmarini:

Below you will find the colleague reviews to this paper, along with our response. Reviewer comments are in black. Responses are provided in *blue italics*. Together with the comments and responses we attach a copy of the manuscript text with additional changes tracked. The revised manuscript with figures is provided as a separate file. We hope you find these responses adequate to merit publication.

Sincerely,

Àlex Martí.

**Reviewer #1**

**General comments:**

1. p. 2, line 30-31
There are on-line operational systems, e.g., for dust transport.

*The text has been modified to include "However, ... , the experiences from other fields (e.g. on-line models for air quality, dust, etc.)..."*

2. p. 10, line 15 and elsewhere in the manuscript The phrase "... decreases with coupling frequency..." may be misunderstood to mean the opposite of what is actually meant, i.e. "... decreases as the coupling frequency increases ..." Therefore, I would recommend to say "... decreases with decreasing coupling frequency ..." instead.

*We agree with the reviewer. We have modified the manuscript with: "decreases with longer coupling intervals..."*

**Suggestions for technical corrections**

1. p.1,line20 Replace "credited to" by "due to"

*Corrected.*

2. p.1,line20 Replace "that" by "as"

*Corrected.*

3. p. 2, line 14 and elsewhere in the text Leave out "of" in "require of"

*Corrected.*

**Reviewer #2**

**General comments:**

1. How does the offline version of your model handle meteorology at the times between the coupling times? On p. 5, line 16, you suggest that meteorological parameters are set to constant in between coupling times. But many offline models linearly interpolate. Could higher-order interpolation schemes reduce error?

*That is a good observation. Thank you. The objective of this paper was to compare the effects of the on-line and off-line coupling strategies between the MetM and VATDM. However, it is true that most off-line systems employed at operational level, perform a linear interpolation in time to attenuate the off-line coupling effects. This is not possible in our off-line strategy because of the concurrent solution of both meteorology and dispersal. Higher-order interpolation schemes to drive the meteorological input would, indeed, reduce the error associated to off-line forecasts.*

2. Section 2.3.1. It was difficult for me to grasp the physical significance of some of the quantities used to compare output from the offline and online models. For example, the Structure component S is said (line 3, page 8) to capture information about the size and shape of cloud objects. But all of the terms in S refer to mass; of a node, column or nodes, or cloud object. How do differences in S reflect variations in size and/or shape?

*Thanks for the comment. The basic idea of the structure (S) component in SAL is to compare the volume of the normalized ash column load (ACL) objects. An object is a group of adjacent grid cells that have an ash cloud loading value above a given threshold. Scaled masses ($V_n$) are calculated separately for each object between the off-line and on-line forecasts. The scaled mass provides the ACL area-integrated (vertical integration for all z nodes) of each forecast, and therefore offers insights of the size and shape of the ACL for the on-line and off-line forecasts.*

The exact meaning of some other parameters, such as R_xy and D were unclear.

*D corresponds to the domain area of the simulation, while R_xy is the area-integrated ACL in grid cell xy within each (on vs off-line) simulation. The text has been updated to clarify this section.*

And it was unclear to me how one could get a value less than zero for L_1.

*That is correct. We apologize for the typo. The values of L_1 are in the range of [-1,1], with L_1=0 suggesting identical centers of mass for both forecast. Corrected.*

Also, I didn't see a definition of the parameter B in eq. 13

*The definition of the parameter B was missing. Thank you! We have added them in the manuscript.*

3. Several source parameter terms in Table 2 are not adequately explained. Details are in comments in that table. Also, I don't see sources cited for the observations used to constrain the eruption source parameters. Other key observations, like the arrival time of the Cordon Caulle ash cloud in Buenos Aires on 4 June 2011 (p. 20, line 3) do not cite sources.

*Thank you. We have included additional information to most parameters in Table 2 and added the corresponding references regarding the arrival of ash to Buenos Aires airports (Collini et al., 2013)*

*Collini, E., Osores, M. S., Folch, A., Viramonte, J., Villarosa, G. and Salmuni, G.: Volcanic ash forecast during the June 2011 Cordón Caulle eruption, Nat. Hazards, 66, 389–412, doi:10.1007/s11069-012-0492-y, 2013.*

4. Some of the figures need more description. For example, the methods of estimating mass eruption rate plotted in Fig. 5b. And the various lines in Fig. 10b representing mass eruption rate with time.

*We have added additional information to each figure's description.*

*Figure 5b. Resulting MER over time considering different plume parameterizations (FPLUME - Folch et al. (2016); Woodhouse - Woodhouse et al. (2013); Mastin - Mastin et al. (2009); Degruyter - Degruyter and Bonadonna (2012));*

*Figure 10b. Resulting MER for each coupling strategy (meteorology coupled on-line or with intervals of time of 1h, 3h, 6h and 12h) with Degruyter option only.*

5. Section 4.1.2: In examining the Eyjafjallajökull eruption, why do you think the misses were in the south and the false alarms were in the north? Dacre et al. (2011) suggested there was an error in the modeled wind speed over England. Was the south-oriented wind just accelerating, and the acceleration was not being caught when the coupling intervals were too infrequent?

*That was an excellent observation. While the errors suggested by Dacre et al (2011, 2016) could be a factor to the errors shown in Fig. 8, the errors in the plume position shown by the off-line forecast are probably caused by the cumulative effect of errors in the infrequent coupling of driving meteorology en route. The synoptic meteorological situation over South Iceland and the North Sea increases (see Fig1. Folch et al., 2012) suggests that the south-oriented wind speed (>30m/s) increased during the 15 and 16 of April (purple area if Fig. 1 from Folch et al., 2012). This scenario, together with employing infrequent intervals to drive the meteorological conditions, could explain why our results showed misses in the south and false*

*alarms in the north of the plume. The manuscript has been updated to clarify this section.*

*Folch, A., Costa, A., Basart, S., 2012. Validation of the FALL3D ash dispersion model using observations of the 2010 Eyjafjallajökull volcanic ash clouds. Atmospheric Environment 48, 165e183.*

*Dacre, H. F., et al. (2011), Evaluating the structure and magnitude of the ash plume during the initial phase of the 2010 Eyjafjallajökull eruption using lidar observations and NAME simulations, Journal of Geophysical Research: Atmospheres, 116(D20),*

*Dacre, H. F., N. J. Harvey, P. W. Webley, and D. Morton (2016), How accurate are volcanic ash simulations of the 2010 Eyjafjallajökull eruption?, J. Geophys. Res. Atmos., 121, 3534–3547, doi:10.1002/2015JD024265.*

6. Section 4.1.3: You say that bias scores suggest that offline forecasts tend to systematically underestimate ash column loading? Is there a physical explanation for this? If the model is conserving mass, does this imply that offline models also systematically overestimate cloud area? And how does this statement square with your statement on p. 19, line 6, that all off-line forecasts OVER-estimate ash column loading?

*Thank you for the observation. First of all, we have noticed that statements claiming overestimation of ACL were incorrect. In most cases, off-line forecast tend to UNDER-estimate ACL compared to the on-line forecast. For the Amplitude component of SAL and the Bias score, off-line forecasts underestimate the domain-averaged ACL when A or Bias are < 0. However, in the case of the Frequency Bias (FBI) score, off-line forecasts underestimate the domain-averaged ACL for values of FBI < 1. The original text was incorrect in the description of some of the FBI and A scores and it has been updated accordantly.*

*The fact that most off-line forecasts show a FBI < 1 at the end of the simulation indicates that the forecast system has a tendency to underestimate ACL events. However, it does not measure how well the off-line forecast corresponds to the on-line simulation, only measures relative frequencies.*

*Amplitude scores, on the other end, provide a simple measure of the quantitative accuracy of the total concentration of airborne ash in the domain ignoring the field's subregional structure. According to Wernli et al. (2008) Amplitude scores range between [-2,2] with 0 denoting no difference between off-line and on-line forecasts. An amplitude score of +1/-1 indicates that off-line forecasts overestimate/underestimate the domain-averaged ACL by a factor of 3. Scores of A=0.4 and 0.67 correspond to factors of 1.5 and 2, respectively. Our results indicate that most off-line forecast have a small tendency (i.e. about 0.2 times) to underestimate ACL objects from the on-line forecasts. The infrequent coupling of the meteorology in the off-line forecasts could be the cause of this difference. In some cases, the infrequent coupling interval (e.g. 3h, 6h, 12h) of an off-line forecast*

*might imply that some of the ash is deposited on the ground or leaves the computational domain. We must highlight that Amplitude scores only compare airborne ACL events within the computational.*

*Wernli, H., Paulat, M., Hagen, M. and Frei, C.: SAL—A Novel Quality Measure for the Verification of Quantitative Precipitation Forecasts, Mon. Weather Rev., 136(11), 4470–4487, doi:10.1175/2008MWR2415.1, 2008.*

Additional minor and specific comments are included in the attached pdf. I look forward to seeing the final version of this paper.

[revised manuscript text omitted]

**Alex Marti 16/10/2017 15:52**

**Comment [1]:** As mentioned by Reviewer 2, this is an advantage when the volcanic cloud is affecting the meteorological parameters. This can be specially important when the meteo conditions change rapidly in time (Marti et al 2017). In addition, Grell and Baklanov, 2011; Zhang, 2008 have listed the several other advantages for employing on-line modeling systems, including amongst others:
−Only one grid for both the Met and scalar concentration advection-diffusion solver.
−No interpolation in space and time.
−Identical physical parameterizations and numerical schemes, i.e., no inconsistencies.
−Availability of all 3-D meteorological variables at the right time, i.e., each time step.
−Possibility to consider feedback mechanisms, e.g., aerosol forcing.
−No need for meteorological pre/postprocessors.

**Alex Marti 11/10/2017 16:25**

**Alex Marti 19/10/2017 11:18**

**Alex Marti 11/10/2017 17:54**

**Alex Marti 16/10/2017 13:18**

**Alex Marti 11/10/2017 17:54**

**Alex Marti 11/10/2017 17:54**

[revised manuscript text omitted]

Alex Marti 11/10/2017 19:36

Alex Marti 11/10/2017 19:37

Alex Marti 11/10/2017 19:38

Alex Marti 16/10/2017 15:53
**Comment [2]:** Text has been added above to answer the reviewer comment.

Alex Marti 11/10/2017 19:40

[revised manuscript text omitted]

Alex  Marti 17/10/2017 11:26
**Comment [3]:** Scaled masses ($V_n$) are calculated separately for each object between the off-line and on-line forecasts.  The scaled mass provides the ACL area-integrated (integral of z nodes) of each forecast, and therefore provides insights of the size and shape of the ACL for the on-line and off-line forecasts.

Alex  Marti 16/10/2017 17:08
**Comment [4]:** This is the vertical integration of the ACL fields in 2D (xy).

Alex  Marti 16/10/2017 17:21

values ($R$). This provides a simple measure of the quantitative accuracy of the total concentration of ash in the domain ignoring the field's subregional structure:

$$A = \frac{\bar{R}_{off} - \bar{R}_{on}}{0.5|\bar{R}_{off} + \bar{R}_{on}|} \quad (4)$$

where $\bar{R}_{off}$ and $\bar{R}_{on}$ are the ash masses averaged over all grid cells in the domain, i.e., $\bar{R}_n = \sum_{(xy)\in Domain} R_{xy} / domain\ area$. Amplitude scores range between [-2,2] with 0 denoting no difference between off-line and on-line forecasts. An amplitude score of +1/-1 indicates that off-line forecasts overestimate/underestimate the domain-averaged ACL by a factor of 3. Scores of $A = 0.4$ and 0.67 correspond to factors of 1.5 and 2, respectively (Wernli et al., 2008).

The location ($L$) component of SAL compares the mass distribution between forecasts. The $L$ component is composed by two parts:

$$L = L_1 + L_2 \quad (5)$$

The first one [$L_1$, Eq. (6)] compares the normalized distance between the center of mass ($C$) of the off-line and on-line ACL fields over the maximum distance within the entire domain ($d$):

$$L_1 = \frac{|C_{off} + C_{on}|}{d} \quad (6)$$

The values of $L_1$ are in the range of [0,1], with $L_1 = 0$ suggesting identical centers of mass for both forecast. However, separated ash clouds could also have the same center of mass, and therefore $L_1 = 0$ would not necessary indicate a perfect match. The second part of the location component [$L_2$, Eq. (7)] aims at distinguish such situations by measuring the weighted average [$H$, Eq. (8)] between the center of mass of the total ash load and the center of mass for each object ($C_n$):

$$L_2 = 2\left[\frac{|H_{off} + H_{on}|}{d}\right] \quad (7)$$

Alex Marti 16/10/2017 17:34
Comment [5]: Considers airborne ash in the computational domain only. None of the SAL components account for ash outside of the domain or ash deposited in the ground.

Alex Marti 16/10/2017 17:38

Alex Marti 16/10/2017 17:37

Alex Marti 16/10/2017 17:36

Alex Marti 16/10/2017 17:39

Alex Marti 12/10/2017 08:27

[revised manuscript text omitted]

Alex Marti 16/10/2017 12:55
**Comment [7]:** That was a good observation. While the errors suggested by Dacre et al (2011, 2016) could be a factor to the errors shown in Fig. 8, the errors in the plume position shown by the off-line forecast are probably caused by the cumulative effect of errors in the infrequent coupling of driving meteorology en route. The synoptic meteorological situation over South Iceland and the North Sea increases (see Fig1. Folch et al., 2012) suggests that the south-oriented wind speed (>30m/s) increased during the 15 and 16 of April (purple area). This scenario, together with employing infrequent intervals to drive the meteorological conditions could explain why our results showed misses in the south and the false alarms in the north of the plume.

Folch, A., Costa, A., Basart, S., 2012. Validation of the FALL3D ash dispersion model using observations of the 2010 Eyjafjallajökull volcanic ash clouds. Atmospheric Environment 48, 165e183.

Dacre, H. F., et al. (2011), Evaluating the structure and magnitude of the ash plume during the initial phase of the 2010 Eyjafjallajökull eruption using lidar observations and NAME simulations, Journal of Geophysical Research: Atmospheres, 116(D20),

Dacre, H. F., N. J. Harvey, P. W. Webley, and D. Morton (2016), How accurate are volcanic ash simulations of 
[revised manuscript text omitted]

Alex Marti 16/10/2017 13:18